# Vascular permeability in retinopathy is regulated by VEGFR2 Y949 signaling to VE-cadherin

**Ross O Smith[1], Takeshi Ninchoji[1], Emma Gordon[1†], Helder André[2], Elisabetta Dejana[1,3], Dietmar Vestweber[4], Anders Kvanta[2], Lena Claesson-Welsh[1]***

[1]Department of Immunology, Genetics and Pathology, Rudbeck Laboratory, Science for Life Laboratory Uppsala University, Uppsala, Sweden; [2]Department of Clinical Neuroscience, Division of Eye and Vision, St. Erik Eye Hospital, Karolinska Institutet, Stockholm, Sweden; [3]IFOM-IEO Campus Via Adamello, Milan, Italy; [4]Max Planck Institute for Molecular Biomedicine, Münster, Germany

**Abstract** Edema stemming from leaky blood vessels is common in eye diseases such as age-related macular degeneration and diabetic retinopathy. Whereas therapies targeting vascular endothelial growth factor A (VEGFA) can suppress leakage, side-effects include vascular rarefaction and geographic atrophy. By challenging mouse models representing different steps in VEGFA/ VEGF receptor 2 (VEGFR2)-induced vascular permeability, we show that targeting signaling downstream of VEGFR2 pY949 limits vascular permeability in retinopathy induced by high oxygen or by laser-wounding. Although suppressed permeability is accompanied by reduced pathological neoangiogenesis in oxygen-induced retinopathy, similarly sized lesions leak less in mutant mice, separating regulation of permeability from angiogenesis. Strikingly, vascular endothelial (VE)-cadherin phosphorylation at the Y685, but not Y658, residue is reduced when VEGFR2 pY949 signaling is impaired. These findings support a mechanism whereby VE-cadherin Y685 phosphorylation is selectively associated with excessive vascular leakage. Therapeutically, targeting VEGFR2-regulated VE-cadherin phosphorylation could suppress edema while leaving other VEGFR2-dependent functions intact.

***For correspondence:**
lena.welsh@igp.uu.se

**Present address:** [†]Institute for Molecular Bioscience, The University of Queensland, St Lucia, Australia

**Competing interests:** The authors declare that no competing interests exist.

## Introduction

Blood vessel dysfunction in the retina is a critical component of several blinding eye diseases. In particular, the retina is sensitive to the fluid accumulation that can result from excessively leaking blood vessels in the eye and reduction of the ensuing edema is a therapeutic goal in the clinic (*Daruich et al., 2018*). Vascular endothelial growth factor A (VEGFA) is one of the key factors controlling endothelial cell biology in the eye (*Daruich et al., 2018*). The importance of VEGFA signaling is underscored by the clinical success of antibodies (bevacizumab, ranibizumab) neutralizing VEGFA, or soluble receptor fusion proteins (aflibercept), neutralizing several VEGF family members. These drugs preserve vision in many patients, but over time disease progression may resume, and for others there is no benefit even from the initial anti-VEGF treatment (*Bressler et al., 2017*). The role of VEGFA for survival of VEGFR2-expressing retinal neurons presents an additional challenge (*Nishijima et al., 2007*). Moreover, anti-VEGF treatment has been linked to capillary loss and rarefaction of normal adult vasculature (*Kamba and McDonald, 2007*). While a focus in the development of new therapies to treat retinopathies has been on increasing the potency of anti-VEGFA therapy, the detrimental effects of long-lasting VEGFA suppression are now being recognized (*Usui-*

**eLife digest** The number of people with impaired vision and blindness is increasing in Western society due to the aging population and the increased prevalence of diabetes. This has led to eye diseases, such as age-related macular degeneration and diabetic retinopathy becoming more common. In both these eye diseases, new blood vessels grow in the retina – the light-sensitive part of the eye – to bring oxygen and nutrients to the tissue. However, these new blood vessels are leaky and allow molecules to leave the bloodstream and enter the retinal tissue. This causes the retina to swell and impair a person's vision. The leaky blood supply also reduces the amount of oxygen that gets to the tissue, resulting in further damage to the retina.

When tissues experience low levels of oxygen, cells start making a protein called vascular endothelial growth factor (or VEGF for short). Whilst this protein is important for helping form new blood vessels, it also makes these vessels leaky. Current treatments for age-related macular degeneration and diabetic retinopathy decrease swelling in the eye by blocking the action of VEGF. However, these treatments also cause existing blood vessels and nerve cells to die, leading to irreversible damage. Now, Smith et al. have set out to find whether the effects of VEGF can be blocked without causing further damage to existing cells.

To investigate this possibility, the eyes and retinas of mice were treated with a laser or exposed to changing oxygen levels to create injuries that resembled human age-related macular degeneration and diabetic retinopathy. Each of the tested mice had specific mutations in proteins known to interact with VEGF. Fluorescent particles were injected into the bloodstream of the mice to assess how these different mutations affected blood vessel leakage: if fluorescent particles could no longer be detected outside the blood vessels, this suggested that the mutation had stopped the vessels from leaking. Further experiments showed these specific mutations affected leakage and did not prevent new blood vessels from forming.

In the future it will be important to see if drugs, rather than mutations, can also decrease the leakiness of blood vessels in the retina. Such chemical compounds could then be tested in mouse experiments. If successful, these drugs might be used to treat patients with age-related macular degeneration and diabetic retinopathy.

*Ouchi and Friedlander, 2019*). Taken together, the development of alternative treatments to block excessive vascular leakage in the eye while sparing other VEGF functions is very important.

The retina is protected by a blood retinal barrier (BRB) which, under normal conditions, limits the extravasation of blood components into the tissue. The BRB consists of an inner barrier at the level of retinal vessels and an outer barrier at the cell junctions of the retinal pigment epithelium and it serves to stringently protect the tissue from pathogens, edema and inflammation (*Zhao et al., 2015*). The BRB is established in conjunction with vessel entry in the central nervous system during early development (*Chow and Gu, 2017*). In eye diseases such as diabetic retinopathy, hypoxia and the ensuing increased production of VEGFA is accompanied by breakdown of the BRB (*Campochiaro, 2015*). The molecular mechanisms downstream of VEGFA leading to this disruption have not been defined.

VEGFA, originally identified as vascular permeability factor (VPF) (*Senger et al., 1986*), binds to VEGFR2 inducing receptor phosphorylation and the propagation of signaling cascades regulating endothelial survival, proliferation and motility (*Simons et al., 2016*). Acute responses to VEGFR2 activity include destabilization of endothelial junctions leading to increased permeability. Notably, phosphorylation of VEGFR2 at Y949 (951 in human) enhances permeability via the disruption of adherens junctions in the dermal vasculature, but also in pathologies such as neuroendocrine cancer, melanoma and glioblastoma (*Li et al., 2016a*). Phosphorylation of Y949 creates a binding site for the adaptor molecule T cell specific adaptor (TSAd), which in turn binds c-Src at endothelial junctions (*Li et al., 2016a*; *Matsumoto et al., 2005*; *Sun et al., 2012*). c-Src is implicated in phosphorylation of vascular endothelial (VE)-cadherin (*Li et al., 2016a*; *Adam et al., 2010*; *Vestweber, 2008*; *Wallez et al., 2007*). VE-cadherin is the main component of endothelial adherens junctions, forming homophilic interactions between endothelial cells, and is critical in regulation of permeability in response to VEGFA as well as inflammatory cytokines (*Li et al., 2016a*; *Akla et al., 2018*;

*Lampugnani et al., 2018*). Tyrosine phosphorylation of VE-cadherin is associated with different biological functions. Thus, phosphorylation of Y658 and Y685 is associated with the control of vascular permeability (*Orsenigo et al., 2012*), and angiogenic sprouting (*Gordon et al., 2016*), and Y658 with junction stability (*Garrett et al., 2017*; *Schulte et al., 2011*), whereas phosphorylation of Y731 in VE-cadherin is linked to leukocyte extravasation (*Wessel et al., 2014*). Serine phosphorylation of VE-cadherin at S665 is also implicated in regulation of endothelial junctions (*Gavard and Gutkind, 2006*).

Here, the molecular mechanisms underlying vessel leakage in eye diseases were studied using mouse models deficient in the VEGFR2-TSAd-VE-cadherin pathway. Interruption of the VEGFR2 pY949F pathway resulted in suppressed vascular leakage at the choroid or at the superficial retinal vasculature in mouse models of age-related macular degeneration and diabetic retinopathy, respectively. Furthermore, disrupting this pathway correlated with reduced VE-cadherin pY685 levels and reduced pathological neoangiogenesis of the superficial retinal vasculature. These data identify VEGFR2 pY949 signaling as an important contributor to edema in retinopathies which may serve as a basis for development of new therapies selectively suppressing VEGFA-dependent disruption of the vascular barrier.

## Results

### Reduced vessel leakage from $Kdr^{Y949F/Y949F}$ retinopathy models

The $Vegfr2^{Y949F/Y949F}$ mouse (henceforth denoted $Kdr^{Y949F/Y949F}$), lacks the pY949 phosphosite of VEGFR2 and therefore fails to induce activation of c-Src. We have previously shown that $Kdr^{Y949F/Y949F}$ exhibits suppressed vessel permeability in the dermis specifically in response to VEGFA (*Li et al., 2016a*). The stringent BRB of the retinal vasculature is not expected to be regulated by VEGFA, however, in ocular disease such as retinopathy, the BRB may be broken down (*Urias et al., 2017*); in accordance, proliferative retinopathies are characterized by increased transvessel flow and edema (*Campochiaro, 2015*; *Kim et al., 2016*; *Luo et al., 2011*; *Stahl et al., 2010a*). We therefore set out to determine whether the pY949 signaling pathway regulates pathologic leakage in the setting of proliferative retinopathy.

To induce retinopathy, $Kdr^{Y949F/Y949F}$ mice and their wild-type littermates (henceforth referred to as $Kdr^{+/+}$) were exposed to a laser-induced chorodial neovascularization (CNV) model. The vascular lesions established after laser pulse disruption of the Bruch's membrane mimic the progression of exudative age-related macular degeneration (*Lambert et al., 2013*). The lesions develop from the choroidal vessels over the course of several days as a consequence of hypoxia and elevated production of VEGFA (*André et al., 2015*). Choroids of CNV-treated $Kdr^{+/+}$ and $Kdr^{Y949F/Y949F}$ mice were collected 14 days post-laser injury, a timepoint corresponding to a phase of completed neoangiogenesis and relative maturation of vessels lesions (*André et al., 2015*). Before collection, vessel leakage from the lesions was examined by monitoring extravasation of circulating 100 nm fluorescent microspheres, a particle size selected as being the smallest that would not simply leak through fenestrated pores of the choroid (*Gupta et al., 2015*). After 2 min of circulation, the microspheres remaining in circulation were flushed away by cardiac perfusion and choroid tissue was collected, immunostained, and analyzed by confocal microscopy for lesion size and microsphere accumulation. Lesions were of equal size (*Figure 1A–B*; 42038 µm² ± 2514, $Kdr^{+/+}$; 44399 µm² ± 3454, $Kdr^{Y949F/Y949F}$), still, microsphere accumulation was significantly reduced in the vicinity of $Kdr^{Y949F/Y949F}$ lesions as compared to the $Kdr^{+/+}$ lesions (*Figure 1C–D*). We conclude that pathological leakage in the choroid can be suppressed by attenutation of pY949 VEGFR2 signaling and that this decrease is not simply due to an anti-angiogenic effect of interrupted VEGFA signaling.

To extend this finding we applied another common retinopathy model, oxygen-induced retinopathy (OIR). In the OIR model, mice are exposed to 75% oxygen during postnatal (P) days 7–12 after which they are returned to normal atmosphere (21% oxygen). During the first stage, VEGFA expression is suppressed which leads to apoptosis of capillaries in the central region of the retina (*Lange et al., 2009*). In the second stage, retinal VEGFA expression is induced as a consequence of moving mice from high oxygen to normal atmosphere (relative hypoxia), leading to the formation of pathological neovascular tufts, characterized by being disorganized and leaky (*Connor et al., 2009*). To determine the role of VEGFR2 pY949 signaling in regulating leakage from neovascular tufts, 25

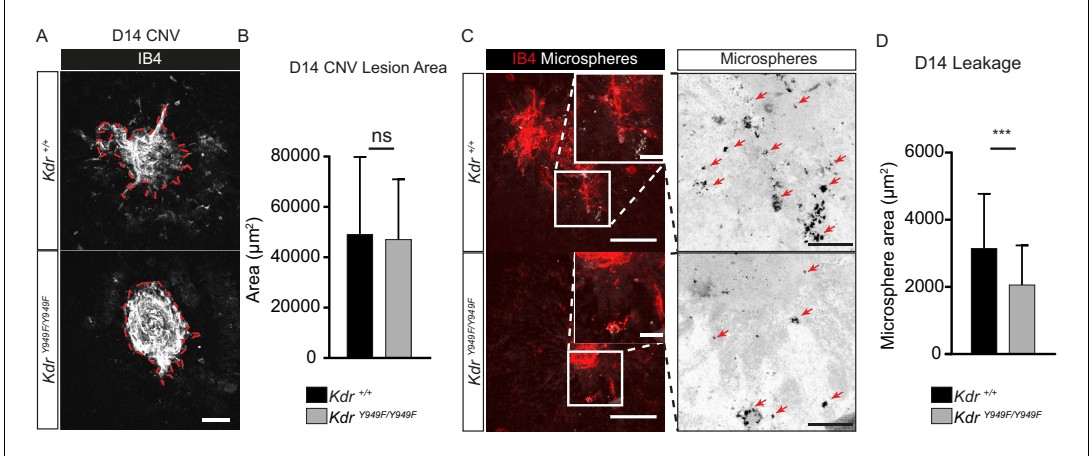

**Figure 1.** Reduced leakage from CNV lesions in $Kdr^{Y949F/Y949F}$ retinas at D14. (A) Representative CNV lesions imaged from whole mount choroid tissue, collected at day (D) 14 after laser injury, from $Kdr^{Y949F/Y949F}$ and $Kdr^{+/+}$ littermates, immunostained for isolectin B4 (IB4). Scale bar = 100 µm. Dotted red line highlights the extent of lesion formation. (B) Quantification of average lesion size at D14 after injury. n = 60–67 lesions per group from 9 to 11 mice per group. ns = not significant p=0.6882. (C) Representative images of D14 lesions from $Kdr^{Y949F/Y949F}$ and $Kdr^{+/+}$ littermates immunostained for IB4 (red), showing accumulation of tail-vein injected, fluorescent 100 nm microspheres (white) in the tissue around the lesions. Insets enlarged (right) with microspheres shown as black dots on white background. Scale bar = 100 µm. Inset scale bar = 25 µm. Arrows point to areas of microsphere accumulation. (D) Quantification of the average area of accumulated microspheres per image after 2 min of circulation. n = 35–74 lesions per group from 7 to 14 mice per group. ***p<0.001 p=0.0006.

The online version of this article includes the following source data for figure 1:

**Source data 1.** Text file containing the ImageJ macro code used to quantify microsphere accumulation in CNV experiments.
**Source data 2.** Excel file containing the collected CNV lesion size and microsphere area.

nm fluorescent microspheres, a particle size chosen as the smallest commercially available, were injected in the tail vein of P17 pups, and allowed to circulate for 15 min before the remaining microspheres were flushed away by cardiac perfusion.

As expected, retinas from P17 $Kdr^{+/+}$ pups submitted to OIR had a considerable central avascular region remaining following the early phase of vessel destruction and also abundant retinal neovascular tuft formation. Of note, at P17, $Kdr^{Y949F/Y949F}$ mice had a reduction in tuft formation compared to the $Kdr^{+/+}$ mice (**Figure 2A–C**; see **Figure 2—figure supplement 1A** for images without overlay). This was not due to differences in body weight of P17 pups (**Table 1**). The reduced tuft formation was a result of suppressed pathological angiogenesis during the P12-17 period when mice were kept in normal atmosphere following the hyperoxia period. At P12, the morphology of the retina vasculature was similar for $Kdr^{Y949F/Y949F}$ and $Kdr^{+/+}$ littermates (**Figure 2—figure supplement 1B–C**).

Importantly, extravascular accumulation of microspheres in regions of neovascular tuft growth was significantly suppressed in $Kdr^{Y949F/Y949F}$ retinas compared to $Kdr^{+/+}$ (**Figure 2D–E**). Note that vessel-proximal microspheres, sticking to the either side of the wall of the disorganized tufts, were not included in the quantification. We conclude that disrupting pY949 signaling during OIR leads to a suppression of pathological vessel leakage and also to a reduction in pathological neoangiogenesis. The decrease in tuft formation in itself results in reduced overall leakage, however, it should be noted that remaining tufts leaked less in OIR-challenged $Kdr^{Y949F/Y949F}$ mice (**Figure 2E**). Moreover, the extent of infiltration of inflammatory CD68+ and CD45+ cells were similar at P17 for the $Kdr^{Y949F/Y949F}$ and $Kdr^{+/+}$ strains indicating that the enhanced barrier in the $Kdr^{Y949F/Y949F}$ pups was specific for macromolecules and did not exclude inflammatory cells (**Figure 2—figure supplement 2A–C**).

## pY949 signaling axis results in altered VE-cadherin phosphorylation

The canonical signalling pathway downstream of the pY949 phosphosite consists of the Src Homology 2 (SH2) containing adaptor-protein TSAd (gene designation $Sh2d2a$), which binds to pY949 with

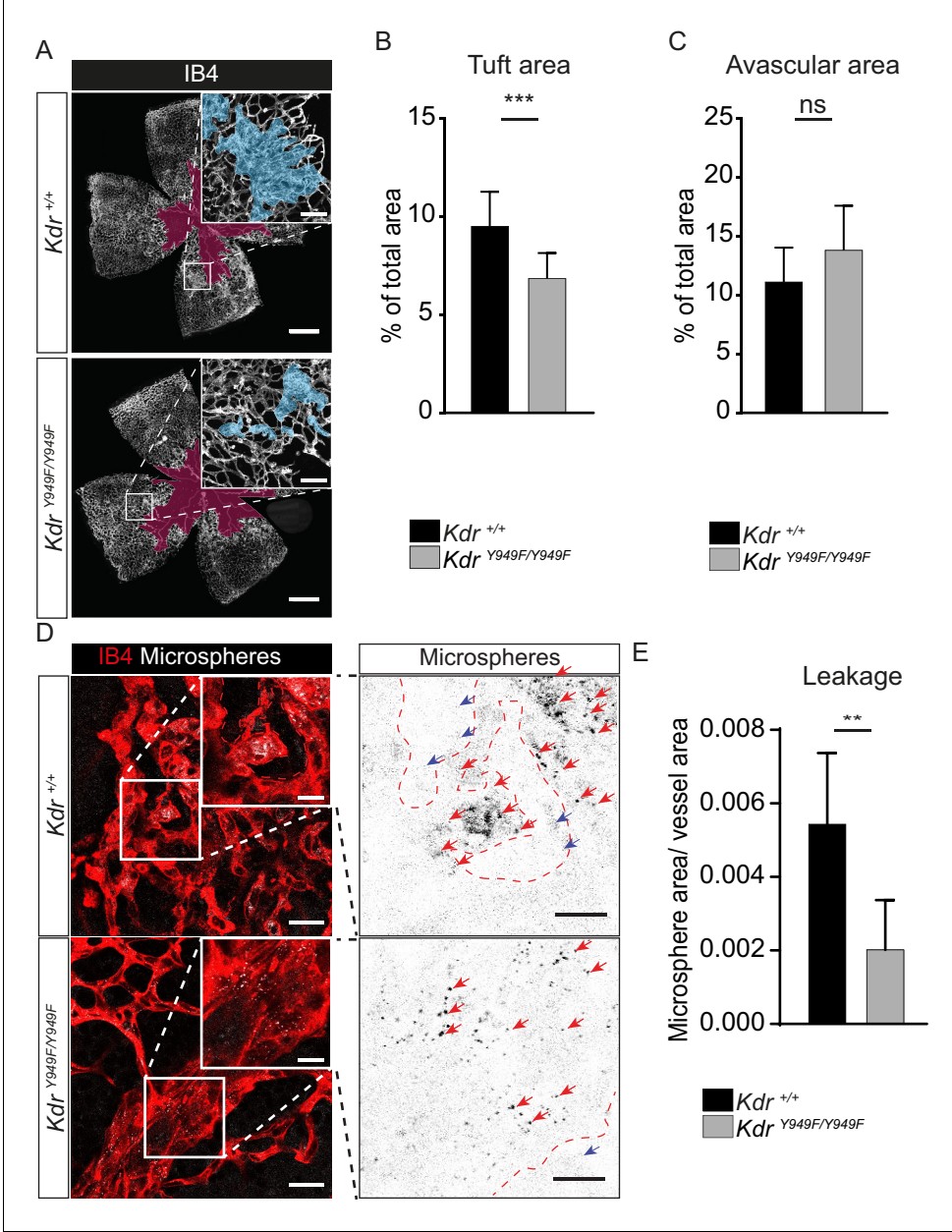

**Figure 2.** Reduced leakage from OIR lesions in $Kdr^{Y949F/Y949F}$ retinas at P17. (**A**) Representative images of whole mount retinas from $Kdr^{Y949F/Y949F}$ and $Kdr^{+/+}$ mice, collected on postnatal day (P)17 after OIR challenge, stained with isolectin B4 (IB4). Avascular tissue in the central retina is marked with purple overlay and neovascular tufts, clusters of disordered vessels, are indicated with blue overlay in the insets. See *Figure 2—figure supplement 1A* for corresponding images without color overlays. Scale bar = 500 µm. Inset scale bar = 100 µm. (**B**) Neovascular tuft coverage as percentage of total retina area. (**C**) Avascular area as percentage of total retina area. n = 10–14 mice, mean value of both eyes;***, p<0.00 p=0.003; ns, not significant p=0.0775. (**D**) Representative images of tufts from $Kdr^{Y949F/Y949F}$ and $Kdr^{+/+}$ mice immunostained for isolectin B4 (IB4;red), showing accumulation of tail-vein injected green-fluorescent 25 nm microspheres (white) in the tissue around the tufts. Insets enlarged (right) with microspheres shown as black dots on white background. Scale bar = 25 µm. Inset scale bar = 10 µm. Dotted line representing the region of IB4 staining. Arrows point to accumulated microspheres; red arrows for microspheres within the IB4 positive region, blue arrows for microspheres away from the vessel wall. (**E**) Quantification of D, showing average area of accumulated extravasated microspheres, normalized to tuft area, per image after 15 min of circulation. n = 6–7 mice per group, 3–11 images per mouse; **p<0.01 p=0.0033. The online version of this article includes the following source data and figure supplement(s) for figure 2:

*Figure 2 continued on next page*

*Figure 2 continued*

**Source data 1.** Text file containing the ImageJ macro code used to quantify microsphere accumulation in OIR experiments.
**Source data 2.** Excel file containing the collected *Kdr* tuft area, avascular area, and extravasated microsphere area.
**Figure supplement 1.** Retina vasculature following OIR.
**Figure supplement 2.** Inflammatory cells in retina tufts following OIR.

high affinity. TSAd in turn recruits c-Src, leading to the phosphorylation of VE-cadherin (illustrated in *Figure 3A*; *Li et al., 2016a*; *Sun et al., 2012*). To assess the role of these downstream players in retinopathy, we expanded our studies using mice treated with tamoxifen at P12 to specifically delete TSAd expression in endothelial cells, $Sh2d2a^{fl/fl}$; *Cdh5-CreERT2* (denoted $Sh2d2a^{iECKO}$). After OIR challenge of $Sh2d2a^{iECKO}$ mice along with their wild-type littermates ($Sh2d2a^{fl/fl}$; referred to as $Sh2d2a^{iECWT}$), tuft formation was reduced in $Sh2d2a^{iECKO}$ mice to a similar extent as seen with $Kdr^{Y949F/Y949F}$ mice (*Figure 3B–C*: see *Figure 3—figure supplement 1A* for images without overlay) while the avascular area was similar in the $Sh2d2a^{iECKO}$ and $Sh2d2a^{iECWT}$ mice at P17 (*Figure 3D*). Cre recombination in the $Sh2d2a^{iECKO}$ model was approximately 80% as assessed by vessel area (*Figure 3—figure supplement 1B,C*), in agreement with immunoblotting analysis performed earlier to show the efficiency of *Sh2d2a* excision in this model (*Gordon et al., 2016*). The distribution of recombined endothelial cells was similar between neoangiogenic tufts and the normal vasculature.

The cytoplasmic tyrosine kinase c-Src phosphorylates VE-cadherin in vitro (*Adam et al., 2010*; *Orsenigo et al., 2012*) and its activity correlates with detection of phosphorylated VE-cadherin in vivo (*Orsenigo et al., 2012*). Immunostaining for pY418, a tyrosine residue in c-Src kinase domain required for maximal kinase activity, revealed active c-Src at the periphery of the retina and elevated levels in the neovascular tufts of $Kdr^{Y949F/Y949F}$ and $Kdr^{+/+}$ mice (*Figure 3E*). Of interest, some pY418 c-Src immunostaining was localized at cell-cell junctions as indicated by co-staining with VE-cadherin. Notably, the level of junctional pY418 c-Src was similar in tufts from $Kdr^{Y949F/Y949F}$ and $Kdr^{+/+}$ mice (*Figure 3F*), indicating that pY949 signaling does not account for c-Src activity at cell-cell junctions in this model. Thus, signaling through pY949 in VEGFR2 appeared not to contribute to regulation of c-Src kinase activity at endothelial junctions in the retina vasculature.

As VE-cadherin is the major component of adherens junctions and thus vital for vessel integrity (*Giannotta et al., 2013*), we performed immunostaining of P17 $Kdr^{Y949F/Y949F}$ and $Kdr^{+/+}$ retinas after OIR, to examine VE-cadherin expression and junction morphology. Neovascular tufts exhibited junctions with an irregular VE-cadherin morphology compared to non-tuft regions (*Figure 3G,H*, compare VE-cadherin morphology in tuft and non-tuft panels showing $Kdr^{+/+}$), suggestive of increased internalization and degradation of VE-cadherin in the tufts (*Bentley et al., 2014*; *Dejana et al., 2008*). In agreement, tuft endothelial junctions were detected by antibodies against two known phosphorylation sites in VE-cadherin, pY658 and pY685 (*Figure 3G,H*), associated with

**Table 1.** Body weight at P17 of mice subjected to OIR.

| Mouse strain/genotype | Body weight (+ / - SD) |
|---|---|
| $Kdr^{+/+}$ | 6.53 (0.94) |
| $Kdr^{Y949F/Y949F}$ | 6.86 (0.65) |
| $Sh2d2a^{iECWT}$ | 5.75 (0.76) |
| $Sh2d2a^{iECKO}$ | 5.68 (1.02) |
| VEC-Y685F – wildtype littermates | 6.49 (1.02) |
| VEC-Y685F | 6.37 (1.02) |
| VEC-WT – wildtype littermates | 5.83 (0.76) |
| VEC-WT | 5.94 (0.92) |

The online version of this article includes the following source data for Table 1:
**Source data 1.** Excel file containing the bodyweight information for individual mice used in OIR experiments.

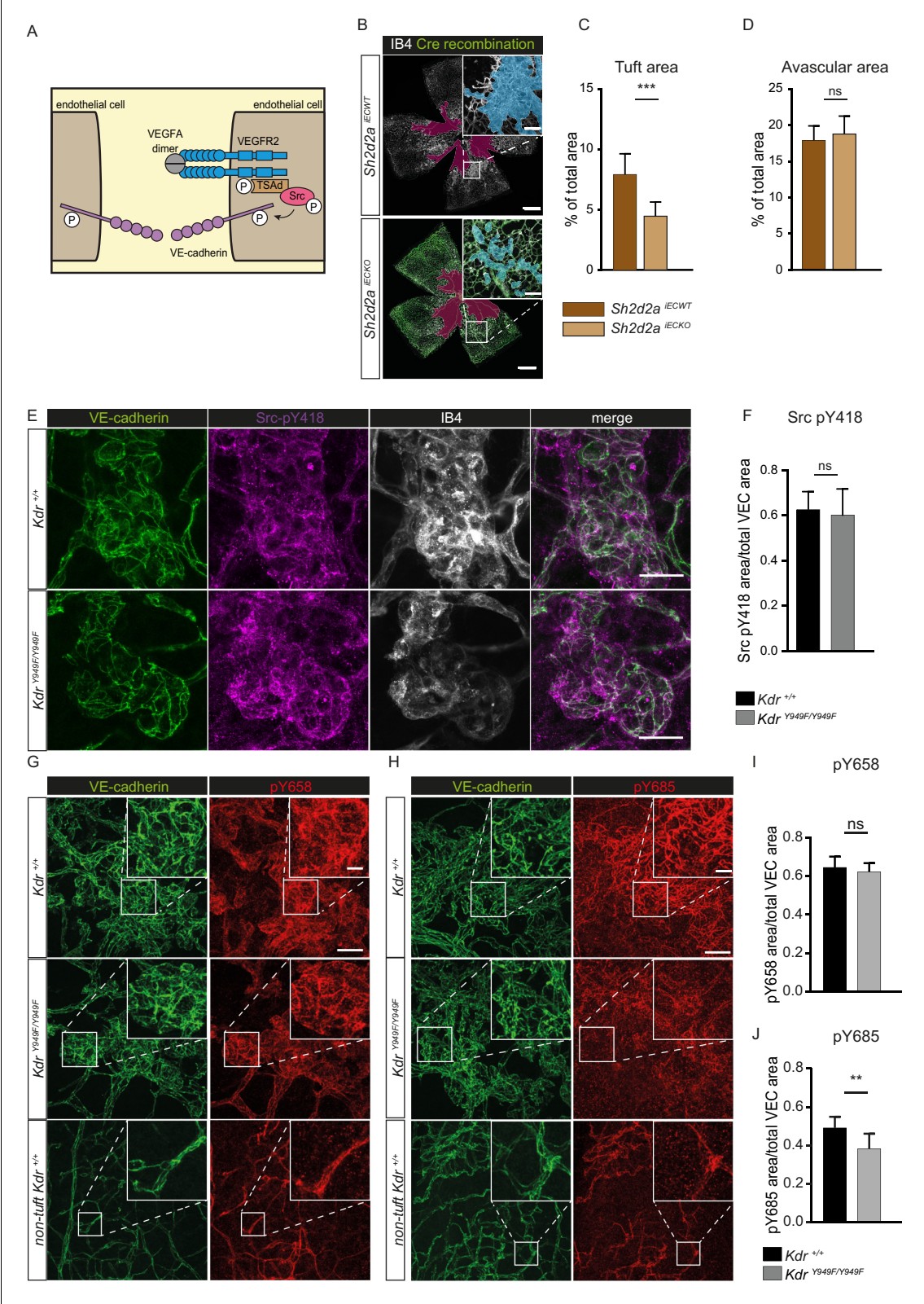

**Figure 3.** pY949 signaling axis involvement in retinopathy pathology. (**A**) Graphic representation of VEGFR2 signaling cascade initiated by Y949 phosphorylation. (**B**) Representative images of whole mount retinas from *Sh2d2a^iECKO^* and *Sh2d2a^iECWT^* mice, collected on postnatal day (P)17 after OIR challenge, stained with isolectin B4 (IB4) with green color marking GFP-positive cells indicating TSAd-deficiency. Avascular area shown with purple overlay, neovascular tufts shown as blue overlay in inset. Scale bar = 500 μm. Inset scale bar = 100 μm. (**C**) Neovascular tuft coverage as percentage of

*Figure 3 continued on next page*

*Figure 3 continued*

total retina area. (**D**) Avascular area as percentage of total retina area. n = 10–15 mice, mean value of both eyes ***, p<0.001 p=0.0001; ns, not significant p=0.3680 E) Representative maximum intensity projections of tufts from $Kdr^{Y949F/Y949F}$ and $Kdr^{+/+}$ mice immunostained for isolectin B4 (IB4; white), VE-cadherin (green), and pY418 c-Src (magenta). Scale bar = 25 µm. (**F**) Quantification shows percentage tuft junctional area, as defined by VE-cadherin immunostaining, positive for pY418 c-Src. n = 7–8 retinas from four mice per group; mean value from four images per retina ns = not significant p=0.6334. (**G–H**) Representative maximum intensity projections of tufts from $Kdr^{Y949F/Y949F}$ and $Kdr^{+/+}$, as well as non-tuft regions from $Kdr^{+/+}$ retinas immunostained for VE-cadherin (green) and G) VE-cadherin pY658 (red) or H) for VE-cadherin pY685 (red). (**I**) Quantification of percentage pY658 immunostaining in tufts, in relation to total tuft junctional (VE-cadherin) area. (**J**) Quantification of percentage pY685 immunostaining as in I. Scale bars in G, H = 50 µm. Inset scale bar = 10 µm. n = 4–6 mice, one retina per mouse, from three independent experiments, 5–9 images per group. ns, not significant p=0.4845, **p<0.01 p=0.0086.

The online version of this article includes the following source data and figure supplement(s) for figure 3:

**Source data 1.** Excel file containing *Sh2d2a* OIR tuft and avascular area, *Kdr* junctional pSrc, and *Kdr* VE-cadherin phosphorylation area.
**Figure supplement 1.** Retina vasculature following OIR in $Shd2da^{iECKO}$ and VE-cadherin phosphorylation in different vessel types.

VE-cadherin turnover (*Orsenigo et al., 2012*). The level of pY658 immunostaining was similar between $Kdr^{Y949F/Y949F}$ and $Kdr^{+/+}$ tufts (*Figure 3I*). In contrast, pY685 levels were significantly lower in the $Kdr^{Y949F/Y949F}$ retinal tufts compared to $Kdr^{+/+}$ (*Figure 3J*).

The underlying vascular plexus showed positive immunostaining with the VE-cadherin phosphoantibodies as well, though to a lesser degree (*Figure 3G,H*). Arteries throughout the retina, including in the neovascularized regions, essentially lacked immunostaining for pY685 and pY658 (*Figure 3—figure supplement 1C*), in keeping with previous findings on the lack of arterial VE-cadherin phosphorylation at these residues (*Orsenigo et al., 2012*).

## Involvement of VE-cadherin pY685 in lesion formation and vessel leakage

To corroborate the role of pY685 VE-cadherin in VEGFR2-regulated neovascular lesion formation, we performed immunostaining for VE-cadherin and pY685 VE-cadherin on choroid tissue from $Kdr^{Y949F/Y949F}$ and $Kdr^{+/+}$ mice after CNV. Lesions at D14 displayed pY685 VE-cadherin immunostaining, though the intensity of the pY685 signal was significantly lower in the $Kdr^{Y949F/Y949F}$ lesions (*Figure 4A,B*). This result suggests that junctional VE-cadherin phosphorylation at Y685 is dependent on pY949 VEGFR2 signaling also in the choroid vasculature.

The importance of VE-cadherin Y685 phosphorylation in the development of retinopathy was further demonstrated using a VE-cadherin Y685F mutant mouse model and its corresponding wild-type construct (VEC-Y685F and VEC-WT, respectively) (*Wessel et al., 2014*), in the OIR model. In these strains, wild-type and mutant *CDH5* cDNA is introduced, replacing the endogenous mouse *Cdh5* gene. The two strains were maintained separately and each therefore had their separate C57Bl/6 wild-type littermates carrying the intact mouse *Cdh5* gene. The extent of oxygen-induced neovascular tuft formation as well as the avascular area in VEC-WT P17 mice was indistinguishable from wild-type C57Bl/6 littermates (*Figure 4—figure supplement 1A–C*) and the avascular area at P12 was likewise indistinguishable for these strains (*Figure 4—figure supplement 1D–E*). At P17, however, tuft formation was reduced in the VEC-Y685F pups compared to VEC-WT controls (*Figure 4C–D*), to a similar extent as seen for the $Kdr^{Y949F/Y949F}$ and $Sh2d2a^{iECKO}$ strains in relation to their respective controls (*Figures 2* and *3*). As for the $Kdr^{Y949F/Y949F}$ and $Sh2d2a^{iECKO}$ strains strains, the avascular area after high oxygen-exposure was equivalent between VEC-Y685F and VEC-WT mice (*Figure 4E*).

Immunostaining with the anti-VE-cadherin phosphoantibodies revealed complete loss of pY685 immunostaining in the VEC-Y685F retinas, validating the specificity of the antibody (*Figure 4—figure supplement 1F*). There was no difference in the levels of total VE-cadherin or VE-cadherin pY658 detected by immunostaining, in the VEC-Y685F tufts compared to WT (*Figure 4—figure supplement 1F*). Neovascular tuft leakage following OIR was examined following the same protocol as above. Microsphere accumulation was significantly reduced in the tufts of VEC-Y685F mice following OIR, compared to VEC-WT (*Figure 4G–H*). As a control, microsphere leakage was assessed in VEC-WT and littermate C57Bl/6 P17 mice after OIR, which showed no difference in extravascular accumulation (*Figure 4—figure supplement 1G–H*). Combined, these data show an essential role for VEGFR2 and downstream VE-cadherin Y685 phosphorylation in elevated vascular leakage in retinopathies.

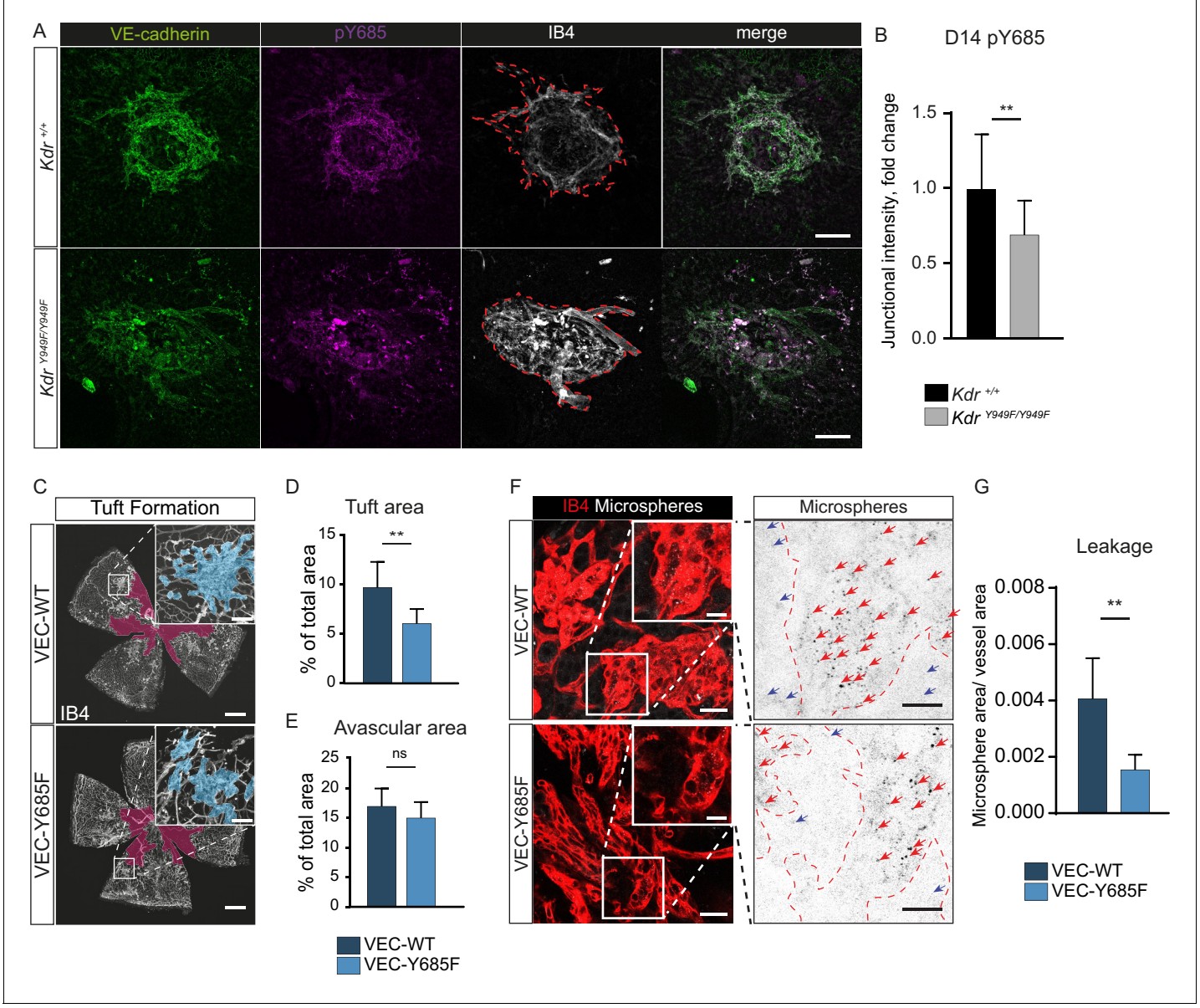

**Figure 4.** Involvement of VE-cadherin pY685 in lesion formation and vessel leakage. (**A**) Representative CNV lesions imaged from whole mount choroid tissue, collected at D14 from $Kdr^{Y949F/Y949F}$ and $Kdr^{+/+}$ littermates, immunostained for VE-cadherin (green), pY685 (magenta) and isolectin B4 (IB4; white). Scale bar = 100 μm; dotted red line highlights the extent of lesion formation in the IB4 channel. (**B**) Quantification of junctional pY685 immunostaining in the lesions. Junctional intensity expressed as the fold reduction of intensity as compared to the average $Kdr^{+/+}$ lesion intensity. n = 14–28 lesions per group from 3 to 5 mice per group, **p<0.01 p=0.0071. (**C**) Whole mount retinas from VEC-Y685F mice and VEC-WT mice, collected on P17 after OIR challenge, stained for IB4. Avascular area shown with purple overlay, neovascular tufts shown as blue overlay in inset. Scale bar = 500 μm. Inset scale bar = 100 μm. (**D**) Tuft coverage and E) avascular area. n = 8–11 mice per group, average of two retinas per mouse. **p<0.05 p=0.0012; ns = not significant, p=0.1535. (**F**) Representative images of accumulation of 25 nm green-fluorescent microspheres (white) in VEC-Y685F and VEC-WT control mice stained for isolectin B4 (IB4; red), showing accumulation in the tissue around the tufts. Insets enlarged (right) with microspheres shown as black dots on white background. Scale bar = 25 μm. Inset scale bar = 10 μm. Dotted line representing the region of IB4 staining. Arrows point to accumulated microspheres; red arrows for microspheres within the IB4 positive region, blue arrows for microspheres away from the vessel wall. (**G**) Quantification of F showing average area of accumulated extravasated microspheres, normalized to tuft area, per image after 15 min of circulation. n = 5–7 mice per group; 10–18 images per mouse **p<0.01, p=0.0016.

The online version of this article includes the following source data and figure supplement(s) for figure 4:

**Source data 1.** Excel file containing collected VE-cadherin pY685 staining in $Kdr$ CNV lesions, VEC-Y685 tuft and avascular area, and VEC-Y685 extravasated microsphere area.

**Figure supplement 1.** VEC-WT and VEC-Y685F mice in OIR.

## Discussion

Here we demonstrate that blocking VEGFA-induced VEGFR2 pY949 signaling at any of several steps in the downstream pathway, leads to reduced vascular leakage in two independent retinopathy models, OIR and CNV. While exposure to high oxygen in the OIR procedure causes pathological vascularization in the superficial retinal vessels, the CNV laser injury targets the choroidal vasculature. Thus, we implicate VEGFR2 pY949 signaling in exaggerated vascular leakage in two very different vascular beds in the retina.

Importantly, we identify components of a signaling axis that can be targeted in order to control excessive vascular leakage and pathological neoangiogenesis without interfering with physiological functions of VEGFA. Although pathological vascular tuft formation was diminished in the $Kdr^{Y949F/Y949F}$ strain, careful characterization of the vasculature in different organs of this strain has not revealed deficiencies in vessel density or morphology during development (*Li et al., 2016a*). Of note, the improved barrier properties in the $Kdr^{Y949F/Y949F}$ vasculature do not exclude infiltration of CD68/CD45+ inflammatory cells (*Li et al., 2016a* and *Figure 2—figure supplement 2*). The effect of bacterial infections and wound healing properties remain to be tested. Nevertheless, based on the currently available information, we favor the notion that suppressed macromolecular leakage in response to VEGFA plays an important role in pathologies while it is dispensable in physiology. We do not exclude that leakage suppression to some extent interferes with neoangiogenesis due to the lack of a fibrinogen mesh for the new vessel to grow on *Dvorak et al. (1995)*. However, as leakage was suppressed in the $Kdr^{Y949F/Y949F}$ retinas in both the CNV and OIR (normalized to neovascular area) models, our data support the concept that leakage is regulated separately from angiogenesis per se. We conclude that in a range of tissues, either in the CNS or in peripheral organs, the VEGFR2 – TSAd – VE-cadherin pathway regulates vascular leakage in the context of disease, independent of angiogenesis.

A considerable strength of our study is in the use of intravenously injected microspheres to assess leakage, as we are able to address both pathological vessel growth and leakage simultaneously. This is superior to traditional methodologies using Evans blue injection, to determine the area of dye leakage over the retina surface, or measure the relative amount of dye extracted from retinas using formamide (*Xu et al., 2001*). Unlike the methodology described here, neither of these methods allow for the precise visualization of the leakage or how it relates to vessel morphology.

Studies in rodent models have suggested that a complete loss of VEGFA-induced signaling in the eye can lead to atrophy of the neural layers, which mechanistically can be explained by the presence of VEGFR2 on retinal neurons (*Okabe et al., 2014*; *Saint-Geniez et al., 2008*). In rats, anti-VEGFA treatment does not lead to damage of neural cells or loss of photoreceptor function (*Long et al., 2018*), indicating that VEGFA might indeed be dispensable for the neural retina in this species. However, retrospective and long term follow-up studies in human patients treated with anti-VEGFA therapies do provide examples of rare negative side effects such as hemorrhages, potentially linked to a systemic reduction of VEGFA (*Avery, 2014*; *Falavarjani and Nguyen, 2013*; *Keir et al., 2017*). Furthermore, anti-VEGFA treatment may contribute to the progression of geographic atrophy due to insufficient vascularization or to the fact that VEGFA may serve as a vital survival factor for the retinal pigment epithelium (RPE) (*Byeon et al., 2010*). Increased treatment frequency in patients receiving ranibizumab or bevacizumab is associated with greater loss of RPE, indicating geographic atrophy (*Enslow et al., 2016*; *Grunwald et al., 2014*; *Rofagha et al., 2013*). The necessity of repeated intraocular injections (*Patel et al., 2013*) for the administration of current anti-VEGFA therapy reduces patient compliance (*Polat et al., 2017*) and results in rare but serious instances of eye damage caused by the injection procedure, such as endophthalmitis or retinal detachment (*Falavarjani and Nguyen, 2013*; *Li et al., 2016b*). Ultimately, the interaction between VEGFR2 pY949/pY951 and TSAd would be an ideal target to specifically block leakage but save other aspects of VEGFR2 biology.

The literature strongly implicates Src family kinases (SFK) in the phosphorylation of tyrosine residues on VE-cadherin, thereby playing an essential role in controlling vessel permeability (*Adam et al., 2010*; *Orsenigo et al., 2012*; *Wessel et al., 2014*; *Trani and Dejana, 2015*). In the context of OIR, direct inhibition of c-Src activation by kinase inhibitors (*Seo and Suh, 2017*; *Werdich and Penn, 2006*), indirect inhibition of c-Src kinase (*Toutounchian et al., 2017*), or suppressed c-Src expression using siRNA (*Zhang et al., 2010*), all result in decreased tuft formation.

Here, we found that neovascular tufts displayed abundant pY418 c-Src at endothelial junctions. Although we may have expected decreased p418 c-Src levels in the $Kdr^{Y949F/Y949F}$ tufts, the levels were similar in the mutant and $Kdr^{+/+}$ littermates. These data indicate that while activation of c-Src alone may be required for induction of permeability and tuft formation, it may not be sufficient to induce these events. In agreement, a triggering event, in addition to c-Src activity, is needed for opening of paracellular junctions (*Adam et al., 2010*; *Adam et al., 2016*). Also, Orsenigo and cow-orkers detected constitutive, flow-dependent pY418 c-Src in leakage-permissive venules, but described an additional triggering event required for leakage to become established (*Orsenigo et al., 2012*). Our data suggests the scenario that VEGFR2 pY949 signaling leads to close proximity and complex formation between VEGFR2 and VE-cadherin potentially involving c-Src (*Li et al., 2016a*), which may be activated in a VEGFR2-independent manner. A caveat in the inter-pretation of these data is that the pY418 antibody used to detect activated c-Src may cross-react with the related Yes and Fyn antibodies. Currently, reagents specifically recognizing only c-Src are missing.

The acute stimulation of blood vessels with bradykinin or histamine to induce leakage is linked to a strong and transient drop in VE-cadherin phosphorylation due to internalization and degradation (*Orsenigo et al., 2012*). We find that in the chronic disease models studied here, regions of greater VE-cadherin phosphorylation displayed increased leakage. The decrease in leakage observed in the $Kdr^{Y949F/Y949F}$ mice challenged with OIR or CNV, correlated with a reduction in VE-cadherin pY685 immunostaining, identifying this phosphosite as a key mediator in leakage regulation. This finding is in line with the observation that sites of leakage in a cremaster model correlated with staining for VE-cadherin pY685 (*Wessel et al., 2014*). The mechanism underlying the important role of VE-cad-herin pY685 compared to pY658 in vascular permeability is not understood. The pY685 phosphosite serves as the binding site for C-terminal Src Kinase (CSK), a negative regulator of c-Src activity. CSK phosphorylates c-Src at pY527, leading to its inactivation (*Latour and Veillette, 2001*). Reduced pY685 VE-cadherin levels, as seen in the $Kdr^{Y949F/Y949F}$ mice, would predict less CSK at the junction and ultimately increased c-Src activity (*Baumeister et al., 2005*; *Ha et al., 2008*), which we did not observe. Moreover, CSK deletion in vitro does not alter barrier properties (*Adam et al., 2010*). Another important aspect of the role of VE-cadherin pY685 is its regulation of pathological angio-genesis. Notably, tuft area was decreased per se, in the VEC-Y685F mutant (*Figure 3B,C*). VE-cad-herin has been implicated in regulation of VEGFR2 signaling by limiting receptor internalization. In the presence of VE-cadherin, VEGFR2 is dephosphorylated by density-enhanced phosphatase (DEP) 1, thus decreasing the levels of active VEGFR2 and proliferative signaling (*Lampugnani et al., 2006*). We hypothesize that VE-cadherin pY685 downstream effectors are critical in VEGFR2 signaling at junctions.

The VE-cadherin Y658 site has also been identified as an integral regulator of junctional stability. Phosphorylation of the site acts to displace bound p120 catenin leading to destabilized adherens junctions (*Garrett et al., 2017*; *Schulte et al., 2011*). VE-cadherin Y658 is phosphorylated by Src family kinases, with low levels of shear stress leading to maximal pY658 immunostaining (*Orsenigo et al., 2012*; *Conway et al., 2017*). In the OIR model, tufts are known to have perturbed perfusion, which may drive the high pY658 intensity in tufts, to a similar extent in the WT and mutant models, as observed in this study.

At this point, VEGFA-targeting is the only pharmacological strategy exploited clinically and it is therefore of particular interest to understand how edema is established in response to VEGFA. Other molecular regulators of pathological leakage have been described that may operate in con-cert with or independently of the VEGFR2 Y949 signaling. Semaphorin 3A (*Cerani et al., 2013*), Neuropilin 1 (*Fantin et al., 2017*), activin-like kinase receptor type I (*Akla et al., 2018*), angiopoie-tin-like 4 (*Babapoor-Farrokhran et al., 2015*), VE-PTP and Tie2 (*Frye et al., 2015*; *Shen et al., 2014*) have all been implicated in the control of junctional integrity in different vascular beds. Here, we provide information on the specific VEGFR2 signaling pathway leading to edema that can be tested as a readout in other systems. We also present an effective methodology to quantify leakage in relation to pathological angiogenesis, which we foresee will aid in exploration of new targets to treat retinopathies.

# Materials and methods

## Key resources table

| Reagent type (species) or resource | Designation | Source or reference | Identifiers | Additional information |
|---|---|---|---|---|
| Strain, strain background (*Mus musculus*) | *Kdr*$^{Y949F/Y949F}$ | DOI: 10.1038/ncomms11017 | | C57Bl/6 background |
| Strain; strain background (*Mus musculus*) | *Sh2d2a*$^{fl/fl}$; *Cdh5-CreERT2* | DOI: 10.1126/scisignal.aad9256 | | C57Bl/6 background |
| Strain; strain background (*Mus musculus*) | *mT/mG* | DOI: 10.1002/dvg.20335 | | C57Bl/6 background |
| Strain; strain background (*Mus musculus*) | VEC-Y685F | DOI: 10.1038/ni.2824 | | C57Bl/6 background |
| Strain; strain background (*Mus musculus*) | VEC-WT | DOI: 10.1038/ni.2824 | | C57Bl/6 background |
| Antibody | Rat anti-VE-cadherin | BD Biosciences | Catalogue no: 555289 RRID:AB_395707 | (1:100) |
| Antibody | Rabbit anti-VE-cadherin pY658 | DOI: 10.1038/ncomms2199 | | (1:50) |
| Antibody | Rabbit anti-VE-cadherin pY688 | DOI: 10.1038/ncomms2199 | | (1:50) |
| Antibody | Rabbit anti-phospho-Src (Tyr418) | Thermo Fisher Scientific | Catalogue no: 44–660G RRID:AB_1500523 | (1:100) |
| Antibody | Goat anti-CD45 | BD Biosciences | Catalogue no: 553076 RRID:AB_394606 | (1:300) |
| Antibody | Rat anti-CD68 | BioRad Laboratories | Catalogue no: MCA1957 RRID:AB_322219 | (1:300) |
| Antibody | Donkey anti-Rat | Thermo Fisher Scientific | Catalogue no: A-21208 RRID:AB_141709 | (1:500) |
| Antibody | Donkey anti-Rabbit | Thermo Fisher Scientific | Catalogue no: A-31572 RRID:AB_162543 | (1:500) |
| Antibody | Donkey anti-Goat | Thermo Fisher Scientific | Catalogue no: A-21432 RRID:AB_2535853 | (1:500) |
| Software | ImageJ | NIH, Bethesda, MD, USA | RRID:SCR_003070 | |
| Software | GraphPad Prism | GraphPad | RRID:SCR_002798 | |
| Other | Alexa Fluor 488-Isolectin B4 | Thermo Fisher Scientific | Catalogue no I21411 RRID:AB_2314662 | (1:500) |
| Other | Alexa Fluor 594-Isolectin B4 | Thermo Fisher Scientific | Cataolgue no: I21413 RRID:AB_2313921 | (1:500) |
| Other | Alexa Fluor 647-Isolectin B4 | Thermo Fisher Scientific | Catalogue no: I32450 RRID:SCR_014365 | (1:500) |
| Other | Fluoro-Max Dyed Blue Aqueous Fluorescent Particles | Thermo Fisher Scientific | Catalogue no: B0100 | |
| Other | Fluoro-Max Dyed Green Aqueous Fluorescent Particles | Thermo Fisher Scientific | Catalogue no: G25 | |

## Animal studies

Mouse husbandry and oxygen-induced retinopathy (OIR) challenge took place at Uppsala University, and the University board of animal experimentation approved all animal work for those studies. Choroidal neovascularization (CNV) experiments took place at Karolinska Institutet, St. Erik Eye Hospital, Stockholm; the procedures were approved by the Stockholm's Committee for Ethical Animal Research. Animal handling was in accordance to the ARVO statement for the Use of Animals in Ophthalmologic and Vision Research. All animal experiments were repeated at least three independent times with wildtype and mutant mice compared within the same litter when possible. Sample size was chosen to ensure reproducibility and allow stringent statistical analysis. Randomization of mice and blinding of the investigators were not performed. No mice were excluded from analyses, though CNV lesions that fused with neighboring lesions were excluded from analysis.

A knock-in mutation in the VEGFR2 gene, *Kdr*, was created by homologous recombination using VelociGene technology (Regeneron Pharmaceuticals, New York, USA), wherein the tyrosine (Y) at position 949 was replaced with phenylalanine (F) (*Li et al., 2016a*). The *Kdr^Y949F/Y949F* mice, initially on mixed 129S6/C57BL/6 background, were extensively backcrossed to C57BL/6J (Taconic Biosciences). *Sh2d2a^fl/fl*; *Cdh5-CreERT2* mice, referred to as inducible Endothelial Cell-specific Knock Out *Sh2d2a^iECKO* mice (previously denoted *Tsad^iECKO* mice), were generated as described (*Gordon et al., 2016*). Inducible deletion was by intraperitoneal injection of tamoxifen (Sigma, T5648) at P12, upon removal from hypoxia, and 24 hr later at P13 (400 µg/dose). To track recombination, the *Sh2d2a^iECKO* strain was crossed with the *mT/mG* strain (JAX stock #007676). Cre-driven recombination resulted in *mTmG*-dependent conversion to green fluorescent protein (GFP) (*Muzumdar et al., 2007*). VE-cadherin (VEC)-Y685F and VEC-WT mice were generated using either WT or Y685F mutant human *CDH5* (VE-cadherin gene designation) cDNA to replace the endogenous mouse *Cdh5* gene (*Wessel et al., 2014*).

## OIR

A standard OIR model was used as described with minor modification (*Connor et al., 2009*). Briefly, each litter of pups was placed, along with the mother, into a chamber that maintained an oxygen concentration of 75.0% (ProOx 110 sensor and A-Chamber, Biospherix, Parish, NY). Mice remained in the chamber for 5 days, beginning at P7 and extending until P12, when they were returned to normal atmosphere (~21% oxygen) until termination at P17. While the pups remained at 75% oxygen throughout, lactating adult females were removed from the chamber on P8, P9, P10, and P11 for 2 hr a day, to prevent oxygen toxicity-related death. At P17, pups were weighed and sacrificed and eyes enucleated and fixed in 4% paraformaldehyde (PFA) at room temperature for 30 min. For microsphere extravasation experiments, mice at P17 were briefly warmed under a heat lamp to dilate tail veins before a tail vein-injection of microspheres (1% solution of 25 nm green fluorescent polystyrene beads; 50 µl per mouse (ThermoFisher Cat.no. G25). Microsphere size was chosen based on previous experience in analyzing VEGFA-regulated leakage from non-fenestrated vessels (*Li et al., 2016a*). Microspheres were allowed to circulate for 15 min before final sacrifice and tissue collection. To remove blood and microspheres from the retinal vessels, mice were perfused with phosphate-buffered saline (PBS). Mice were fully anesthetized using isofluorane inhalation or alternately by an intraperitoneal injection of a mixture of ketamine/xylazine (100 mg/kg ketamine; 20 mg/kg xylazine), after which room temperature PBS was flushed through the vasculature. Litters of mice exposed to OIR protocol were excluded from analysis when the average weight of pups at P17 was less than 5.5 grams, as low weight may indicate maternal neglect or other reasons for inability to thrive (*Stahl et al., 2010b*). See *Table 1* for average weight of pups at P17.

## CNV

A standard protocol of laser-induced CNV was employed (*Lambert et al., 2013*; *André et al., 2015*). Briefly, 6–14 week-old *Kdr^Y949F/Y949F* mice were anesthetized (ketamine/xylazine; 30 mg/kg, 5 mg/kg respectively) and pupils dilated using a topical administration of tropicamide (0.5%; Alcon, Puurs, Belgium). Choroidal neovascularization lesions were induced in both eyes by diode laser (532 nm; IRIS Medical, Mountain View, CA, USA) with settings: 75 µm spot, 200 mW intensity, 100 ms duration. All visual hemorrhagic lesions were excluded from the study. After laser-induction, animals were treated twice with 1 mL of saline (9 mg/mL NaCl; B. Braun, Melsungen, Germany)

subcutaneously under the back skin to prevent dehydration and the eyes were kept lubricated by topical administration of a paraffin and Vaseline mix (APL, Gothenburg, Sweden). At post-laser day 8, or 14, mice were culled and eyes immediately enucleated. The retina tissue was carefully dissected away to expose the choroid and CNV lesions. At day 14, prior to sacrifice, mice were warmed on heating pads and given a tail vein-injection of microspheres (1% solution of 100 nm Blue-fluorescent polystyrene beads; 100 µl per mouse (ThermoFisher Cat.no. B100) followed by 2 min of circulation and perfusion with fixative via cardiac puncture under isoflurane anesthesia. A larger microsphere size (100 nm) was chosen than for the OIR analyses (25 nm) to avoid spontaneous passage of small microspheres through the fenestrated choroidal vasculature. When the CNV lesions created in one eye grew and fused together, all fused lesions were excluded from further analysis.

## Antibodies

Retinal vasculature and CNV lesions were immunostained with directly conjugated Alexa Fluor 488-Isolectin B4, Alexa Fluor 594-Isolectin B4, or Alexa Fluor 647-Isolectin B4 (Sigma). Endothelial cell junctions and phosphorylated VE-cadherin were stained with anti-VE-cadherin antibody (1:100; BD Rat 555289) and affinity purified rabbit antibodies against VE-cadherin pY658 and pY685 (*Orsenigo et al., 2012*). Phosphorylated c-Src was assessed using anti-phospho-Src (Tyr418) Antibody (1:100; Invitrogen Rabbit 44–660G). Secondary antibodies used were Alexa488 anti-Rat (1:500; Invitrogen Donkey A-21208) and Alexa555 anti-Rabbit (1:500; Donkey A-31572). Inflammatory cells were stained with anti-CD45 (1:300; BD Biosciences Goat 553076) and anti-CD68 (1:300; BioRad Rat MCA1957). Secondary antibodies used were Alexa488 anti-Rat (1:500; Invitrogen Donkey A-21208). Alexa555 anti-Goat (1:500; Invitrogen Donkey A-21432).

## Immunofluorescent staining

Whole mount immunostaining of retinas and choroids was performed following OIR and CNV experiments. Dissected issues, fixed in PFA were first incubated in blocking buffer (1% bovine serum albumin/2% fetal calf serum/0.05% Na-deoxycholate/0.5% Triton X-100/0.02% Na Azide in PBS) for 2 hr to block unspecific binding. Incubation with primary antibodies and secondary antibodies was carried out sequentially over night at 4°C on a rocking platform. Tissues were mounted on slides with Fluormount G mounting media (SouthernBiotech).

Microscopy was done using a Zeiss LSM700 microscope or a Leica SP8 confocal microscope. Images were acquired with the 20x, 40x or 63x objective. Processing and quantification of images was done with ImageJ software (NIH).

## Quantification of avascular area and neovascular tufts

Neovascular tuft formation and avascular area were determined by immunostaining retinas followed by image analysis. Quantification of total vascularized area, central avascular area, and tuft area was performed by outlining images manually in ImageJ. Using a tilescan of the IB4 channel for each whole mounted retina, the freehand selection tool was used to demarcate the vascular front, creating an ROI (region of interest) for the total vascularized area. The freehand selection tool was also used to outline IB4 positive vessels from neovascular tufts – regions with disorganized dilated vessels often with markedly intense IB4 staining. The ROIs for tufts were merged into a single ROI corresponding to the all the neovascular tuft area for a given retina. The tuft area normalized to the total vascularized area was reported as a percentage of the total retina that contained tufts. Similarly, the avascular region was determined using the freehand selection tool to outline the central avascular regions. Regions where the superficial layer of capillaries was absent were determined and merged to form a single ROI corresponding to all of the avascular regions for a given retina. The avascular area normalized to the total vascularized area was reported as a percentage of the total retina that was still avascular. The researcher was blinded to the genotype of the sample when performing quantifications.

## Quantification of fluorescent microspheres in CNV

The quantity of extravasated fluorescent microspheres, marking sites of leakage, was measured in digital fluorescent images of CNV lesions from images taken with a Leica SP8 confocal microscope (63X objective) equipped with single and dual fluorescence filters charge-coupled device (CCD)

camera. Camera settings were constant for images from all groups in each experiment. Using ImageJ software (NIH, Bethesda, MD), a guassian blur filter (sigma 1) was applied to each image and a threshold was applied to the microsphere channel (405 for Blue fluorescence) using the Triangle algorithm. Microsphere area was calculated using the Analyze Particles function with an upper limit of 500 pixels to avoid the inclusion of large staining artifacts.

## Quantification of fluorescent microspheres in OIR

For the OIR model, the quantity of extravasated fluorescent microspheres, marking sites of leakage, was measured in digital fluorescent images of regions of retina over 10 to 15 images in each eye taken with a Leica SP8 confocal microscope (63X objective) equipped with single and dual fluorescence filters charge-coupled device (CCD) camera. Camera settings were constant for images from all groups in each experiment. Using ImageJ software (NIH, Bethesda, MD), the microsphere channel and IB4-vessel channel (488 for Green fluorescence and 647 for IB4) were adjusted with threshold (Huang for Ib4 and Triangle for FITC) for each channel. Extravasated microsphere area was calculated by measuring the signal in the green fluorescence channel after removing any signal contained within the ROI (region of interest) corresponding to the IB4-positive area. The Analyze Particles function was employed to quantify the microspheres. A lower limit of 10 pixels was selected to distinguish the microsphere signal from background noise. The mean area density for each group of mice was calculated from the median value of all images of the eyes of each mouse (*Fuxe et al., 2011*). To quantify leakage based on microscopic images, the amount of tracer extravasation was normalized to blood vessel density. The researcher was blinded to the genotype of the sample when performing quantifications.

## Quantification of inflammatory cells in OIR

Immunostaining for CD68, CD45, and IB4 was performed on retinas from $Kdr^{+/+}$ and $Kdr^{Y949F/Y949F}$ mice. Confocal images were analyzed using ImageJ software for the presence of inflammatory cells within neovascular tufts. For each image, the tuft area was manually outlined using the freehand selection tool and then each channel was thresholded (CD68 with Zen; CD45 with Triangle) and the positive area within these tuft regions was calculated as the percentage of total tuft area. The average of five regions per retina are presented for each animal.

## Statistical analysis

Statistical analysis was performed with GraphPad Prism (GraphPad). An unpaired Student's T test was used to compare means between experimental groups. A Mann Whitney U test was used to compare the medians between experimental groups with similar outcome. All tests were two-tailed and $p < 0.05$ was considered a statistically significant result. Values shown are the mean, with standard deviation used as the dispersion measure. Biological replicates refer to individual mice for OIR experiments and to individual lesions for CNV experiments. Independent experiments refer to experiments done in different days with independently generated material. A statistical method of sample size calculation was not used during study design. For in vivo experiments, we used an average of 6 animals per experiment, with a minimum of 3 (detailed number of animals used is given in figure legends). The investigators were blind to the genotype of the animal for data analysis.

## Acknowledgements

We gratefully acknowledge the expert assistance of Marie Hedlund and Pernilla Martinsson, Uppsala University, and Monica Aronsson, St. Erik Eye Hospital. Critical input by Professor Mike Sapieha, University of Montreal, is gratefully acknowledged. Constructive discussions and advice from Fabrizio Orsenigo, IFOM, and Dr. Elisabet Ohlin Sjöström, SOBI, are much appreciated.

## Additional information

### Funding

| Funder | Grant reference number | Author |
|---|---|---|
| Australian Research Council | DE170100167 | Emma Gordon |
| Vetenskapsrådet | 2015-02375 | Lena Claesson-Welsh |
| Knut och Alice Wallenbergs Stiftelse | 2015.0030 | Lena Claesson-Welsh |
| Knut och Alice Wallenbergs Stiftelse | 2015.0275 | Lena Claesson-Welsh |
| Fondation Leducq | 17 CVD 03 | Lena Claesson-Welsh |
| Fondation ARC pour la Recherche sur le Cancer | AIRC IG 18683 | Elisabetta Dejana |

The funders had no role in study design, data collection and interpretation, or the decision to submit the work for publication.

### Author contributions

Ross O Smith, Conceptualization, Formal analysis, Investigation, Writing - original draft, Writing - review and editing; Takeshi Ninchoji, Formal analysis, Investigation, Methodology, Writing - review and editing; Emma Gordon, Conceptualization, Funding acquisition, Writing - review and editing; Helder André, Resources, Investigation, Writing - review and editing; Elisabetta Dejana, Resources, Funding acquisition, Writing - review and editing; Dietmar Vestweber, Anders Kvanta, Resources, Writing - review and editing; Lena Claesson-Welsh, Conceptualization, Resources, Formal analysis, Supervision, Funding acquisition, Writing - original draft, Writing - review and editing

### Author ORCIDs

Ross O Smith (ID) https://orcid.org/0000-0003-4239-3204
Dietmar Vestweber (ID) http://orcid.org/0000-0002-3517-732X
Lena Claesson-Welsh (ID) https://orcid.org/0000-0003-4275-2000

### Ethics

Animal experimentation: Mouse husbandry and oxygen-induced retinopathy (OIR) challenge took place at Uppsala University, and the University board of animal experimentation approved all animal work for those studies (Permit Number 5.8.18-06789-2018). Choroidal neovascularization (CNV) experiments took place at Karolinska Institutet, St. Erik Eye Hospital, Stockholm; the procedures were approved by the Stockholm's Committee for Ethical Animal Research (Permit Number Dnr 49/15). Animal handling was in accordance to the ARVO statement for the Use of Animals in Ophthalmologic and Vision Research.

### Decision letter and Author response

Decision letter https://doi.org/10.7554/eLife.54056.sa1
Author response https://doi.org/10.7554/eLife.54056.sa2

## Additional files

### Supplementary files

• Transparent reporting form

#### Data availability

All data generated or analysed during this study are included in the manuscript and supporting files. Text files containing the ImageJ macros used for automatic detection of microspheres in Figures 1 and 2 are provided.

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
