## [Decision Letter]

**Acceptance summary:**

This work analyzing the effects of a specific VEGFR2 phosphorylation (tyr949) on vascular permeability in vivo on oxygen-induced retinopathy to be interesting and timely, showing selective effects on vascular permeability. We also find that the revised manuscript has largely addressed the reviewer concerns, and we appreciate that you added the experimental numbers, clarified methods and analysis protocols, and added body weight data for the mice. These findings suggest that specific targeting of VEGFA signaling functions may provide therapeutic benefit in the future.

**Decision letter after peer review:**

Thank you for submitting your article "Vascular permeability in retinopathy is regulated by VEGFR2 Y949 signaling to VE-cadherin" for consideration by *eLife*. Your article has been reviewed by three peer reviewers, including Victoria L Bautch as the Reviewing Editor and Reviewer #1, and the evaluation has been overseen by Jonathan Cooper as the Senior Editor.

The reviewers have discussed the reviews with one another and the Reviewing Editor has drafted this decision to help you prepare a revised submission.

Summary:

The manuscript by Smith et al. investigates the mechanism whereby vascular leak is regulated during pathological retinal angiogenesis. They find that a specific phosphorylation site of VEGFR2 (Y949) is crucial for regulating leak in vivo, and show data that supports that Tsad, which binds to Y949, and specific phosphorylation of VE-cadherin on Y685, but not c-Src, are involved in the signaling to leak downstream of VEGFR2. The strengths of the study are the use of in vivo models and readouts that provide evidence that specific genetic manipulations lead to changes in leak, and the potential impact of the work in providing focus on a specific aspect of VEGFA signaling which could prevent the side effects associated with VEGFA therapy. The lack of differential c-Src phosphorylation may be due to the limitations of the reagents, as the antibodies are thought to cross-react with other Src family kinases. The limitations of the study are associated with the fact that phenotypes in vivo are more difficult to quantify. For example, the changes in pY685 VE-cadherin (Figure 3J) in the Y949 mutant background are significant but not robust; however, the actual mutant (VE-cadherin Y685F) shows a reduction in leak (Figure 4G). The major critique centers around better explanation and/or quantification of the phenotypes.

Essential revisions:

1) Please address assay variability at several levels: Better explanation of quantification assays, and where appropriate increase the n of fields analyzed and/or mice used. Please use body weight to normalize for OIR (see below). Consider reanalysis of microsphere leak for those microspheres clearly outside the vessels.

a) It would strengthen the work if the analysis protocols were more carefully explained. For example, the Materials and methods describe outlining to get areas of vascular, avascular, and tufts, but do not explain how areas where chosen for microsphere or antibody intensity analysis.

b) Microsphere analysis – it is a little difficult to interpret the overlay, but many of the spheres seem to be in or very near the tufts. Do the microspheres "get stuck" rather than wash out with the perfusion? It is also not clear how "microsphere area" is defined. It would strengthen the work to better define the quantification process, and perhaps to quantify the microspheres that are clearly not in the vessels. Likewise the quantification of the antibody intensity staining would benefit from more detail.

c) For OIR model, the body weight of mouse pups should be reported, because pups with better postnatal weight gain have less neovessel formation in OIR (PMID: 21056995). In general, at least n=8-10 mice with 3 different litters are recommended to overcome biologic variability and obtain reliable readout from OIR model. Figure 4C-D (n=4-5 mice per group) of particular concern. Other data (Figures 1D, 2E, 4G would benefit from higher n, and n for animals and fields should be clearly stated in the figure legend).

2) Tsad excision – consider more rigorous quantification of the excision, perhaps by Western blot in the lungs, and association of excision and tufts: The authors provide 2 possible explanations for their findings that TSAD iECKO and WT have similar avascular areas: these include incomplete excision of Tsad and/or that some vessel regrowth is regulated upstream of TSAd in the Y949F mutant mice. The degree of excision of TSAd is shown in a supplementary figure – some additional quantification by Western blot or some other method would be supportive and more convincing. Do the tuft regions have fewer recombined cells? This was not possible to distinguish in the figures provided.

3) We request text edits to discuss possible adverse effects of targeting the Y949 of VEGFR2 in other physiological angiogenesis or VEGFR2-dependent processes.

---

## [Author Response]

Essential revisions:1) Please address assay variability at several levels: Better explanation of quantification assays, and where appropriate increase the n of fields analyzed and/or mice used. Please use body weight to normalize for OIR (see below). Consider reanalysis of microsphere leak for those microspheres clearly outside the vessels.a) It would strengthen the work if the analysis protocols were more carefully explained. For example, the Materials and methods describe outlining to get areas of vascular, avascular, and tufts, but do not explain how areas where chosen for microsphere or antibody intensity analysis.

We have now detailed how areas where chosen for microsphere and antibody intensity analysis in the Materials and methods section, in the Results section, and in the legends to Figures 2 and 4 (text marked in grey).

Microsphere analysis: The quantity of extravasated fluorescent microspheres, marking sites of leakage, was measured in digital fluorescent images of regions of retina over 10 to 15 images in each eye taken with a Leica SP8 confocal microscope (63X objective) equipped with single and dual fluorescence filters charge-coupled device (CCD) camera. Camera settings were constant for images from all groups in each experiment. Using ImageJ software (NIH, Bethesda, MD), the microsphere channel and IB4-vessel channel (488 for Green fluorescence and 647 for IB4) were adjusted with threshold (Huang for IB4 and Triangle for FITC) for each channel. Extravasated microsphere area was calculated by measuring the signal in the green fluorescence channel after removing any signal contained within the ROI (region of interest) corresponding to the IB4-positive area. The Analyze Particles function was employed to quantify the microspheres. A lower limit of 10 pixels was selected to distinguish the microsphere signal from background noise. The mean area density for each group of mice was calculated from the median value of all images of the eyes of each mouse. (see [1]. To quantify leakage based on microscopic images, the amount of tracer extravasation was normalized to blood vessel density. The researcher was blinded to the genotype of the sample when performing quantifications.

Antibody intensity i.e. neovascular tuft formation and avascular area: Neovascular tuft formation and avascular area were determined by immunostaining retinas followed by image analysis. Quantification of total vascularized area, central avascular area, and tuft area was performed by outlining images manually in ImageJ. Using a tilescan of the IB4 channel for each whole mounted retina, the freehand selection tool was used to demarcate the vascular front, creating an ROI (region of interest) for the total vascularized area. The freehand selection tool was also used to outline IB4-positive vessels from neovascular tufts – regions with disorganized dilated vessels often with markedly intense IB4 staining. The ROIs for tufts were merged into a single ROI corresponding to the all the neovascular tuft area for a given retina. The tuft area normalized to the total vascularized area was reported as a percentage of the total retina that contained tufts. Similarly, the avascular region was determined using the freehand selection tool to outline the central avascular regions. Regions where the superficial layer of capillaries was absent were determined and merged to form a single ROI corresponding to all of the avascular regions for a given retina. The avascular area normalized to the total vascularized area was reported as a percentage of the total retina that was still avascular.

b) Microsphere analysis – it is a little difficult to interpret the overlay, but many of the spheres seem to be in or very near the tufts. Do the microspheres "get stuck" rather than wash out with the perfusion? It is also not clear how "microsphere area" is defined. It would strengthen the work to better define the quantification process, and perhaps to quantify the microspheres that are clearly not in the vessels. Likewise the quantification of the antibody intensity staining would benefit from more detail.

We hope the response given above clarifies the methodology. In Figures 2D, 4G and Figure 4 –figure supplement G, we now show color-coded arrows to indicate extravasated microspheres. Red arrowheads indicate microspheres inside vessels while blue indicate microspheres outside vessels. Microspheres appearing to be on the vessel wall may be inside or outside; this cannot be distinguished. In this analysis, they have been counted as inside (see response to question “Do microspheres get stuck…”).

Do the microspheres "get stuck" rather than wash out with the perfusion?

It is quite likely that microspheres get stuck inside tufts perhaps due to the abnormal flow in these vessels. The microspheres may get stuck within the vessel or at the basement membrane. After perfusion (with 8 ml PBS), we can see remaining microspheres in tufts but never in normal vessels. For the analysis of OIR leakage, we therefore focus only on the clearly extravasated microspheres, away from the vessel wall, with any microsphere signal within the IB4 positive vessel area being removed. We have normalized the microsphere area to tuft area and therefore, the different tuft areas between wildtype and the various mutants we analyze do not affect the analyses. The difference between wildtype and mutants instead is due to that the mutants have stabilized junctions and therefore leak less. We have added a comment on the potential stickiness of the tuft vessel wall in the last paragraph of the subsection “Reduced vessel leakage from *Kdr^Y949F/Y949F^* retinopathy models”.

c) For OIR model, the body weight of mouse pups should be reported, because pups with better postnatal weight gain have less neovessel formation in OIR (PMID: 21056995). In general, at least n=8-10 mice with 3 different litters are recommended to overcome biologic variability and obtain reliable readout from OIR model. Figure 4C-D (n=4-5 mice per group) of particular concern. Other data (Figures 1D, 2E, 4G would benefit from higher n, and n for animals and fields should be clearly stated in the figure legend).

We thank the reviewers for these comments which are well taken. We are aware of that the body weights influence the neovessel formation. We have now given the appropriate reference [2] (subsection “OIR”). Body weight of pups is given in Table 1.

Concerning the number of animals; we would like to emphasize that we have done many independent repeats. The litters that we generate from crossing of heterozygous recombinant animals are generally small (3-5 pups distributed in 25% wildtype, 50% heterozygous and 25% homozygous animals), and it may well be that we don’t get any homozygous pups at all. Therefore, to have as many as 8-10 x 3 mice would require very many breeding couples over lengthy periods of time to obtain up to 30 litters with sufficient numbers of wildtype and mutant pups. To have 8-10 x 3 is feasible when working with adult mice where pairing of gender and age is more important than pairing litter mates, which is essential in the OIR model. However, we certainly do our best to follow Good Research Practice (defined as research of high quality, conducted and reported in a truthful way and with respect to important societal values, and that researchers take responsibility for their research and its consequences) and had continued to collect samples and also managed to have some more litters during the revision period. The n has now been increased for Figure 2B, C and E, Figure 3C and D, Figure 4D, E and G, and Figure 4—figure supplement 1H. As a result, the statistics is considerably improved. We thank the reviewers for this request.

The number of animals now is the following:

Figure 1: n is unchanged

Figure 2B, C: n is wt=10, Y949F=14 (was 8-11)

Figure 2E: n=6-7 (was 5-6)

Figure 3C, D: n=10-15 (was 8-11)

Figure 4D, E: VEC-WT n = 8 and for VEC-Y685F n = 11 (was 4-5).

Figure 4G: n=5-7 (was 3-5)

Figure 4—figure supplement 1H: n=5-6 (was 3-5)

2) Tsad excision – consider more rigorous quantification of the excision, perhaps by Western blot in the lungs, and association of excision and tufts: The authors provide 2 possible explanations for their findings that TSAD iECKO and WT have similar avascular areas: these include incomplete excision of Tsad and/or that some vessel regrowth is regulated upstream of TSAd in the Y949F mutant mice. The degree of excision of TSAd is shown in a supplementary figure – some additional quantification by Western blot or some other method would be supportive and more convincing. Do the tuft regions have fewer recombined cells? This was not possible to distinguish in the figures provided.

The tuft regions have a similar distribution of recombined cells as the rest of the retina. We have given a comment on this fact in the subsection “pY949 signaling axis results in altered VE-cadherin phosphorylation”.

In Gordon et al., [3], Figure 3—figure supplement 1B, we provide extensive blotting data on the degree of excision of Tsad in the tamoxifen-treated *Tsad^fl/fl^;Cdh5-CreERT2* mice (about 80%). We used lung lysates for VEGFR2 ip and TSAd blot to specifically analyze excision in endothelial cells. We now refer to the Gordon et al., paper in the subsection “pY949 signaling axis results in altered VE-cadherin phosphorylation”.

Due to problems with generating sufficient number of *Tsad^fl/fl^* animals of the right genotype to redo the blotting exercise (which is always difficult for technical reasons, poor antibodies etc.), we don’t show this again in this paper.

Finally, with the increased n, there is no longer a significant difference in avascular area between the *Vegfr2^Y949F/Y949F^* and its wildtype and therefore, there is no longer a need to comment on *Vegfr2^Y949F/Y949F^* vis-a-vis the similar avascular area for the *Tsad^fl/fl^* strain and its wildtype. Those comments in the text have been removed.

3) We request text edits to discuss possible adverse effects of targeting the Y949 of VEGFR2 in other physiological angiogenesis or VEGFR2-dependent processes.

We have shown that the *Vegfr2^Y949F/Y949F^*mouse exhibits reduced metastatic spread [4], improved heart function after myocardial infarction (Li et al., submitted) and in the current paper, reduced pathological angiogenesis and leakage in retinopathy models. To what extent the VEGFA-regulated vascular barrier contributes to physiology has remained an enigma. Developmental angiogenesis is unaffected by the improved barrier in the Y949F mouse [4]. It remains to be tested whether wound healing or bacterial infections are affected. However, now, we studied the extent of inflammatory cell infiltration in the wildtype and *Vegfr2^Y949F/Y949F^* strains. There was no difference in the degree of CD68+ and CD45+ cell infiltration. These data have been added in a new Figure 2—figure supplement 2. We comment on this result in the subsection “Reduced vessel leakage from *Kdr^Y949F/Y949F^*retinopathy models” and in the Discussion, stating that inflammatory cell infiltration is not affected by the improved junction stability in the *Vegfr2^Y949F/Y949F^* strain. These results suggest that the improved vascular barrier in the strains studied here preferentially acts to suppress molecular leakage. Moreover, we describe the lack of understanding on the role of vascular barrier stability to VEGFA-induced molecular leakage in physiology. See the Discussion.

References:

1) Fuxe J, Tabruyn S, Colton K, Zaid H, Adams A, Baluk P, Lashnits E, Morisada T, Le T, O'Brien S, et al. (2011) Pericyte requirement for anti-leak action of angiopoietin-1 and vascular remodeling in sustained inflammation. Am J Pathol 178: 2897-2909

2) Stahl A, Chen J, Sapieha P, Seaward MR, Krah NM, Dennison RJ, Favazza T, Bucher F, Lofqvist C, Ong H, et al. (2010) Postnatal weight gain modifies severity and functional outcome of oxygen-induced proliferative retinopathy. Am J Pathol 177: 2715-2723

3) Gordon EJ, Fukuhara D, Westrom S, Padhan N, Sjostrom EO, van Meeteren L, He L, Orsenigo F, Dejana E, Bentley K, et al. (2016) The endothelial adaptor molecule TSAd is required for VEGF-induced angiogenic sprouting through junctional c-Src activation. Sci Signal 9: ra72

4) Li X, Padhan N, Sjostrom EO, Roche FP, Testini C, Honkura N, Sainz-Jaspeado M, Gordon E, Bentley K, Philippides A, et al. (2016) VEGFR2 pY949 signalling regulates adherens junction integrity and metastatic spread. Nat Commun 7: 11017